# mRNA Post-Transcriptional Regulation by AU-Rich Element-Binding Proteins in Liver Inflammation and Cancer

**DOI:** 10.3390/ijms21186648

**Published:** 2020-09-11

**Authors:** Dobrochna Dolicka, Cyril Sobolewski, Marta Correia de Sousa, Monika Gjorgjieva, Michelangelo Foti

**Affiliations:** Department of Cell Physiology and Metabolism and Translational Research Center in Oncohaematology, Faculty of Medicine, University of Geneva, 1211 Geneva, Switzerland; Dobrochna.Dolicka@unige.ch (D.D.); Cyril.Sobolewski@unige.ch (C.S.); Marta.Sousa@unige.ch (M.C.d.S.); Monika.Gjorgjieva@unige.ch (M.G.)

**Keywords:** RNA-binding proteins, HuR, tristetraprolin, nonalcoholic steatohepatitis, fibrosis, hepatocellular carcinoma

## Abstract

AU-rich element-binding proteins (AUBPs) represent important post-transcriptional regulators of gene expression. AUBPs can bind to the AU-rich elements present in the 3’-UTR of more than 8% of all mRNAs and are thereby able to control the stability and/or translation of numerous target mRNAs. The regulation of the stability and the translation of mRNA transcripts by AUBPs are highly complex processes that occur through multiple mechanisms depending on the cell type and the cellular context. While AUBPs have been shown to be involved in inflammatory processes and the development of various cancers, their important role and function in the development of chronic metabolic and inflammatory fatty liver diseases (FLDs), as well as in the progression of these disorders toward cancers such as hepatocellular carcinoma (HCC), has recently started to emerge. Alterations of either the expression or activity of AUBPs are indeed significantly associated with FLDs and HCC, and accumulating evidence indicates that several AUBPs are deeply involved in a significant number of cellular processes governing hepatic metabolic disorders, inflammation, fibrosis, and carcinogenesis. Herein, we discuss our current knowledge of the roles and functions of AUBPs in liver diseases and cancer. The relevance of AUBPs as potential biomarkers for different stages of FLD and HCC, or as therapeutic targets for these diseases, are also highlighted.

## 1. Introduction

Hepatic metabolic disorders and cancers account for more than 3.5% of deaths worldwide [1]. The main etiologies for these diseases are viral infections (i.e., hepatitis B or C virus infections), obesity, alcohol consumption, and medication abuse [2]. The classical histopathological spectrum of these liver diseases starts first with fat accumulation in the hepatocytes (steatosis), which can then progress to chronic liver inflammation (steatohepatitis; Figure 1). Inflammation and lipotoxicity then favor the development of hepatic fibrosis, which can worsen and progress to cirrhosis, a pathological state characterized by severe fibrosis, nodules of regenerating and poorly differentiated hepatocytes, and portal hypertension [3,4]. Although cirrhosis is already a life-threatening disease per se, it also represents a high-risk condition of developing hepatocellular carcinoma (HCC) [4] (Figure 1). Most cases of HCC usually arise in a cirrhotic context, but recent evidence indicates that up to 20% of cases develop in the absence of cirrhosis when only steatosis and/or inflammation is present [5]. This is particularly the case in nonalcoholic fatty liver disease (NAFLD), whose incidence is constantly on the rise and is currently >25% worldwide [6]. With the current pandemic of obesity, HCC incidence in a noncirrhotic context is therefore expected to dramatically increase in the future. 

Hepatic cancers are associated with a high degree of genomic instability and multiple gene mutations. However, nongenomic alterations occurring with metabolic disorders and liver inflammation also importantly foster and contribute to cancer onset and progression. Among those nongenomic alterations are epigenetic modifications (e.g., DNA methylation, chromatin methylation, and acetylation) and mRNA post-transcriptional regulation by noncoding RNAs and specific proteins (e.g., microRNAs (miRNAs) and RNA-binding proteins (RBPs)). Of particular interest for the post-transcriptional regulation of mRNAs is their 3’-UTR, which represents an important site of translational regulation by various mechanisms [7]. A widely characterized mechanism involves the binding of miRNAs to the 3’-UTR of target mRNA, which trigger a blockade of mRNA translation or the decay of mRNAs [8]. Herein, we discuss alternative mechanisms regulating mRNA translation that are mediated by a class of RBPs called AU-rich element-binding proteins (AUBPs). AUBPs bind to the 3’-UTR of mRNAs and thereby provoke either their degradation, stabilization, or translational inhibition. Since AUBPs can target a plethora of transcripts, including inflammatory mediators, cell cycle regulators, proto-oncogenes, and tumor suppressors, they exhibit important regulatory functions in inflammation and cancer development in many organs, including the liver. Regarding liver cancers, our discussion focuses mostly on HCC since the role of AUBPs on other types of liver cancers, e.g., benign liver tumors or intrahepatic cholangiocarcinoma (ICC), has been currently not investigated.

## 2. AU-Rich Element-Binding Proteins

AUBPs are regulatory trans-acting proteins that bind to the 3’-UTR of mRNAs and contain typical sequences enriched in AU bases (AU-rich elements (AREs)) [9]. AREs can be found in the 3’-UTRs of up to 8% of human transcripts and have a length of between 50 and 150 nucleotides [9]. Transcripts usually contain multiple copies of these sequences, of which the most prevalent is the AUUUA pentamer, although other motifs are also commonly found [10]. One or more ARE sequences are found in the mRNA of well-characterized inflammatory cytokines (e.g., interferon γ (IFNγ) and tumor necrosis factor α (TNFα)), but also proto-oncogenes (e.g., c-Fos and c-Myc), tumor suppressors (e.g., fibroblast growth factor 21 (FGF21)), growth factors (e.g., vascular endothelial growth factor (VEGF)), and cell cycle regulators (e.g., cyclins A, B1, and D1) [11,12]. To date, more than 20 distinct AUBPs have been discovered [11], which display tissue specificity (Figure 2) and different affinities to ARE motifs.

### 2.1. Functions of AUBPs

AUBPs are a heterogenous group of protein factors, whose primary function is either to stabilize or destabilize mRNA transcripts or to regulate their translation [13] (Figure 3). Among the AUBPs that destabilize mRNAs, the most well-known are tristetraprolin (TTP), butyrate response factor 1 (BRF1), AU-rich element RNA-binding protein 1 (AUF1), and KH-type splicing regulatory protein (KSRP) [14,15,16]. On the contrary, AUBPs such as Hu-antigen R (HuR) and Hu-antigen D (HuD) improve the half-life of the mRNA transcripts to which they bind [11]. Other AUBPs such as T-cell-restricted intracellular antigen-1 (TIA1) are involved in translational silencing [17] (Figure 3). Some of these AUBPs are associated with the cytoplasmic structures responsible for RNA degradation or storage, such as processing bodies (P-bodies) and stress granules (SGs) [13].

P-bodies are cytoplasmic complexes made of RNAs and proteins that are not embedded by intracellular membranes. They are formed as a result of various stimuli, such as stress or inflammation [18]. P-bodies assemble the enzymes required for mRNA decay (e.g., DCP1/2, CAF-1, CCR4, and XRN1), as well as for RNA interference (e.g., Ago/RISC) [18,19]. AUBPs that destabilize mRNAs (e.g., TTP and BRF1) bind to mRNA transcripts and recruit them to the P-bodies, where they are deadenylated, decapped, and degraded [20,21]. However, other AUBPs such as HuR counteract degradation by preventing mRNA recruitment to the P-bodies and subsequent processing by the degradation machinery [22,23] (Figure 3). 

SGs appear to form as a protective mechanism upon experiencing cellular stresses, i.e., heat shock, hypoxia, and oxidative or endoplasmic reticulum (ER) stress [24]. Some AUBPs, such as TIA1, have the ability to bind mRNAs and to recruit them to SGs [20]. SGs are also nonmembranous cytoplasmic foci, which contain the small ribosomal unit and transcription initiation factors such as eukaryotic initiation factor (eIF) 3, eIF4A, eIF4E, eIF4G, and PABP [25], as well as TIA1 or G3BP1 [20]. SGs form under stress conditions mainly through phosphorylation of eIF2 [20]. The mRNAs stored in SGs are protected from proteasomal degradation during cellular stress and can be later released and translated into proteins [20]. This results in transcript stabilization, along with a transient inhibition of translation [20]. Some AUBPs, such as CUG triplet repeat RNA-binding protein 1 (CUGBP1) or CUGBP2, despite not being the central structural proteins required for SG formation, also have the ability to bind and to stabilize transcripts by associating with SGs, transiently impairing their translation [26] (Figure 3). 

### 2.2. Regulation of the Activity of AUBPs

The regulation of the activity of AUBPs is highly complex, occurs at multiple levels, and is often different depending on the cell/organ under consideration.

#### 2.2.1. Temporal and Spatial Regulation of the Activity of AUBPs

Rapid temporal and spatial changes in cellular transcriptomes, such as those that occur during different developmental stages, require efficient mechanisms allowing for rigorous control of transcription and translation. In this regard, the expression of some AUBPs can be activated through stimulation by various mitogens (Figure 4). For instance, insulin, as well as epidermal growth factor (EGF), fibroblast growth factor (FGF), and platelet-derived growth factor (PDGF), stimulate TTP expression in 3T3 cells, while androgens have been shown to induce HuR expression in hepatic HepG2 cells [27,28,29]. Importantly, the mRNA and protein expression of AUBPs do not always correlate with the proteins’ activity and bioavailability, which can be highly dependent on their intracellular localization. The spatial regulation of AUBPs, and hence their activity, may rely on post-translational modifications as exemplified by the CDK1-dependent Ser^202^ phosphorylation of HuR, which impairs its translocation to its active site, i.e., the cytoplasm, in HeLa cells [30]. Furthermore, TTP promotes P-body formation, which facilitates mRNA degradation, upon transforming growth factor (TGF)–β1 stimulation, consistent with the fact that TTP induction may occur through the TGF–β/Smad pathway [31,32]. TTP has also been shown to colocalize with SGs after heat shock, likely to translocate SG-associated mRNA to P-bodies for degradation [33]. Consistent with such a mechanism, a dual interaction of TTP and BRF1 with both SGs and P-bodies has been reported to mediate the translocation of mRNAs between these two entities [34]. Diverse hypotheses involving protein and mRNA shuttling, as well as a potential merge between SGs and PBs, have been raised, but the precise mechanisms driving P-body and SG assembly, as well as their functional relevance, remain to be precisely defined [35,36]. 

#### 2.2.2. Post-Translational Modifications

To exert their functions, AUBPs need to structurally interact with their target mRNAs, and therefore, post-transcriptional modifications are likely to significantly impact their activity. Most of the available information regarding post-transcriptional modifications of AUBPs concerns TTP and HuR [37,38] (Figure 4). In murine macrophages, the phosphorylation of TTP by the MAPK-activated protein kinase 2 (MK2) has been shown to increase its stability, but also to decrease its binding affinity to ARE sites on target mRNAs [22,39]. Likewise, the phosphorylation of HuR on specific serine residues (Ser^318^ or Ser^202^) by kinases such as CDK5 also reduces the binding of HuR to specific mRNAs such as *CCNA1* in U251 glioma cancer cells or in RKO rectal cancer cells [40,41]. Finally, the phosphorylation and/or ubiquitination of AUBPs have also been reported to drive their proteasomal degradation [42,43,44]. Whether phosphorylation events, or other post-transcriptional modifications such as methylation or glycosylation, also impact the localization and activity of other AUBPs remains to be deciphered. 

#### 2.2.3. Competition and Self-Regulation of the Activity of AUBPs

The binding sites of AUBPs on the 3’-UTR sequences of their mRNA targets often overlap with the binding sites for other regulatory elements, e.g., long noncoding RNAs (lncRNAs) or miRNAs (miRNAs) [9]. In addition, for miRNAs, one AUBP can target many different mRNAs, and a single mRNA can also be targeted by several AUBPs (Figure 4). A striking example of this complexity is illustrated by the 3’-UTR of the *Ptgs2* mRNA, which is targeted by numerous AUBPs, including HuR, TIA1, TIA-1-related protein (TIAR), AUF1, CArG-binding factor A (CBF-A), RNA-binding protein 3 (RBM3), heterogeneous nuclear ribonucleoprotein (hnRNP) A3, and hnRNP A2/B1 in RAW 264.7 macrophages [45]. Several reports have addressed the cooperative and/or antagonistic properties of AUBP, mostly focusing on HuR, to regulate the stability of mRNAs [46,47]. For instance, HuR may stabilize its target mRNAs by protecting them from the degradation activities of other AUBPs. Indeed, HuR has been shown to: (i) Decrease PTBP1 binding to the hepatitis C viral RNA, resulting in higher virus replication [48]; (ii) compete with TTP binding to the *CPB2* mRNA and AUF1 binding to the *ATF3* in HepG2 cells [49,50]; (iii) cooperate with AUF1 to regulate *Mat1A* and *Mat2A* mRNA expression in a rat model of HCC [51]; (iv) compete with CUGBP2 for binding to the *PTGS2* mRNA in cancer cells such as colorectal HT-29 cells [52,53,54]. Depending on the cell/organ, the interaction between different AUBPs can lead also to different outcomes, as illustrated by TIA1 and HuR, which compete in breast cancer cells for binding to the *PDCD4* [55], but which cooperate in bone marrow-derived macrophages (BMDMs) to inhibit the translation of the *Tnf* mRNA [56]. 

Finally, the mRNA of AUBPs can also contain AU-rich motifs in their 3’-UTR, therefore allowing other AUBPs, or themselves, to regulate the stability and translation of their transcript. This is the case for TTP, which has been shown to contain three AUUUA motifs in its 3’-UTR and to exert a negative feedback regulation on its own transcript in murine macrophages RAW 264.7 and in THP-1 human monocyte cells [57,58]. HuR can not only stabilize its own transcript, but can also compete with TTP for binding to it in HEK293 cells [59]. Several other AUBPs, including KSRP, HuR, AUF1, ILF3 (NF90), TIA1, and TIAR, have been further shown to exert self- and cross-regulation of the stability of their transcripts in HeLa cells [60]. 

#### 2.2.4. Interactions with Noncoding RNAs 

miRNAs, lncRNAs, and circular RNAs (circRNAs) are part of the large family of noncoding RNAs (ncRNAs), which have the ability to bind to and to regulate the expression of mRNAs, mostly by restraining their translation or priming them for degradation [61]. In addition to directly acting on their target mRNAs, ncRNAs have been shown to significantly interfere with the activity of AUBPs by either (i) competing for target binding, (ii) facilitating AUBP binding to their target, (iii) destabilizing the mRNAs of AUBPs, or (iv) indirectly inducing post-translational modifications of AUBPs (Figure 4).

In this regard, the lncRNA MEG3, which contributes to hepatic insulin resistance and fibrosis, has been shown to facilitate PTBP1-dependent decay of the *Shp* mRNA, both in hepatic cell lines (Huh7 and Hepa-1) and in vivo [62]. Similarly, the binding of the lncRNA AWPPH to the Y-box-binding protein 1 (YB-1) improved the YB-1-dependent translational activation of the *SNAI1* transcript in SMMC-7721 cells, thereby promoting the prooncogenic phenotype of these cells [63]. On the other hand, ncRNAs can also destabilize the mRNAs of key regulatory proteins of AUBPs under physiological or pathological conditions. This is typically illustrated by the regulation of TTP by the lncRNA Linc-SCRG1 in LX2 stellate cells [64], or of RBM38 by the lncRNA HOTAIR in HCC cells [65]. Numerous miRNAs are also bioinformatically predicted to target AUBPs, with some of them having been experimentally validated in specific conditions, i.e., HNRNPA1 targeting by miR-22 [66], CUGBP2 targeting by miR-95 [67], and YB-1 targeting by miR-148a [68]. 

Finally, indirect mechanisms regulating the activity of AUBPs can also be under the control of lncRNAs or circRNAs. Examples of such indirect regulation are illustrated by (i) the lncRNA highly upregulated in liver cancer (HULC), which triggers YB-1 phosphorylation by the extracellular signal-regulated kinase (ERK) mitogen-activated protein (MAP) kinases, thereby impairing its binding to mRNAs [69]; (ii) the lncRNA MIR22HG, which binds to HuR and prevents its translocation to the cytoplasm and its binding to oncogenes (e.g., *CTNNB1*, *CCNB1*, *HIF1A*, *PTGS2,* and *FOS*) in SMMC-7721 cells [70]; (iii) the lncRNA Ptn-dt, which sequesters HuR from miR-96, thus affecting the stability of this miRNA [71]; (iv) the circRNA circBACH1, which binds to HuR and regulates its translocation into the cytoplasm in Hep3B and HepG2 cells [72]. 

## 3. AUBPs in Liver Inflammation and Fibrosis

Acute hepatic inflammation and its progression toward chronic liver injury require the synergetic action of several regulators of gene expression, including RNA-binding proteins such as AUBPs in different hepatic cell types (e.g., hepatocytes, hepatic stellate cells (HSCs), Kupfer cells, and other immune cells) [64,73]. Of note, the role of AUBPs in immune cell response under general pathological settings has been extensively reviewed in previous publications [74,75]. Particularly relevant to liver diseases, several cytokines, and other proinflammatory molecules, e.g., IL-8, IL-10, IL-6, COX2, TNFα, and GM-CSF, have been shown to harbor ARE-binding motifs in their 3’-UTR that allow rapid regulation by AUBPs [74]. Among all AUBPs, TTP, HuR, and TIA1 appear to play preponderant roles in the modulation of inflammatory processes. TTP has indeed been reported to contribute to the termination of inflammatory responses, since its phosphorylation prevents the destabilization of proinflammatory-related and apoptotic genes in activated macrophages and neutrophils [75]. HuR silencing in activated T cells has been shown to impair Th17 commitment and function in a model of autoimmune encephalomyelitis, but in myeloid cells, HuR silencing potentiates inflammation by overactivating macrophages, thus indicating a dual and cell-specific role for HuR in inflammatory processes [74]. Finally, similarly to TTP, TIA1 knockout mice are more prone to develop inflammation associated with allergic reactions and age-related arthritis, a phenotype associated with the dysregulation of several proinflammatory molecules in activated macrophages, T cells, and NK cells [74].

In the liver, the expression of several AUBPs is strongly altered in patients suffering from nonalcoholic steatohepatitis (NASH) and fibrosis, suggesting that AUBPs are key actors in these pathological processes. However, disparate information is available about single AUBP functions in the liver, and most of the studies have focused only on HuR, TTP, and KSRP.

Regarding HuR, it has been shown to be increased in human fibrotic livers, as well as in mice who underwent bile duct ligation and CCl_4_ treatment [76,77]. The upregulation of HuR appears to sustain liver inflammation and fibrosis by fine-tuning cellular responses in different cells, including HSCs, BMDMs, bone marrow-derived stem cells (BMSCs), and hepatocytes. HuR is indeed involved in BMDM recruitment and macrophage infiltration into the liver upon CCl_4_ treatment by stabilizing an important mediator of liver fibrosis and inflammation, the cannabinoid receptor 1 [77]. Under similar conditions, HuR also mediates BMSC migration to liver injury sites by stabilizing the mRNA of sphingosine 1-phosphate receptor 3 in these cells [76]. The activation of HSCs also appears to be under the control of HuR [78]. Indeed, HuR silencing in activated HSCs has been shown to reduce fibrosis by attenuating inflammation, oxidative stress, and lowering α-smooth muscle actin (α-SMA) and collagen expression. Conversely, PDGF stimulation of HSCs leads to increased cytoplasmic HuR and its binding to the *MMP9*, *ACTB*, *CCL2*, *CCND1,* and *CCNB1* mRNAs [78]. Decreased HSC proliferation/migration has also been observed upon HuR silencing, but TGF–β1 stimulation surprisingly inverts this effect, highlighting the different roles of HuR in the activation pathways of HSCs [78]. Such a paradoxical contribution of HuR to the fibrotic signaling pathways, in particular, is further supported by the observation that HuR induction by sorafenib treatment after bile duct ligation in mice promotes HSC ferroptosis and attenuates fibrosis in vivo [79]. Finally, HuR levels have been shown to correlate positively with liver transplant prognosis and stabilization of the heme oxygenase 1 mRNA in hepatocytes, suggesting that HuR upregulation in this process may represent an important mechanism of protection post-transplantation [80]. The role and therapeutic potential of targeting HuR in liver diseases have been recently and extensively reviewed elsewhere [81].

Regarding TTP, this AUBP has been reported to regulate general inflammatory processes, mainly by controlling the mRNA expression of several cytokines and chemokines (e.g., IL-17, TNFα, IL-6, IL-1β, and CXCL1-2) in distinct cell types (e.g., macrophages, T cells, fibroblasts, and endothelial cells) [82,83,84,85]. TTP downregulation is associated with liver pathologies characterized by inflammation, such as viral infection, hepatic fibrosis, and HCC, but its specific functions in nonimmune cells during hepatic inflammation prior to the development of fibrosis are still unclear. Despite the lack of mechanistic insight, it has been shown that mice depleted of TTP, specifically in the liver, are more resistant to monocyte infiltration upon acute diethylnitrosamine (DEN) injection [86]. In contrast to the role of HuR in activated HSCs, TTP overexpression in LX2 cells not only impairs the cell activation, proliferation, and migration induced by TGF-β treatment, but also induces apoptosis through destabilization of MMP-2 and TNFα [64].

KSRP has been shown to destabilize genes such as *Nos2*, *Ptgs2*, and *Cx3cl1* in gastrointestinal or hepatic epithelial cells following lipopolysaccharide (LPS) or IFN-γ stimulation [87,88]. KSRP was also recently suggested to be involved in the development of insulin resistance (IR), type 2 diabetes (T2D), and NAFLD [89,90]. Indeed, KSRP knockout mice have been shown to develop less steatosis following a high-fat diet (HFD) feeding due to the overexpression of *Per2*, which leads to the disruption of hepatic circadian rhythms [90]. KSRP levels are also significantly downregulated in rats with IR/T2D submitted to a HFD and streptozotocin injection [89].

Scarce information is available about other AUBPs in liver diseases. For instance, AUF-1 and CUGBP1/2 have been described to modulate the stability of the early response genes also targeted by TTP, i.e., *Cxcl8*, *Tnfa, Ptgs2*, and *Csf2* [74], thus indicating again that specific AUBPs can either have synergistic effects or can counteract each other’s actions to balance the cytokine expression during inflammation [84]. YB-1 has been reported to induce hepatic fibrosis due to uninterrupted TGF–β/Smad signaling, which induces continuous HSC activation and extracellular matrix (ECM) production [91]. YB-1 overexpression in LX2 cells is further linked to increased collagen production and α-SMA induced by *Smad2* stabilization. YB-1/*Smad2* interaction has been confirmed upon liver injury induced by CCl_4_ injection [91]. In hepatitis C virus (HCV)-infected Huh7 cells, TIA1 silencing has been reported to interfere with viral RNA replication and infectivity, possibly due to the impairment of HCV-induced SG formation and translational blockage of the interferon-induced transcripts MxA and USP18 [92]. Other AUBPs, e.g., YB-1, TARDBP, HNRNPC, and PTPB1, have also been shown to affect hepatotropic viral replication/infection, but their impact on hepatic inflammation is unclear. For instance, YB-1 is implicated in the stabilization of several HCV protein transcripts [93], whereas TARDBP is suggested to support the hepatitis B virus (HBV) life cycle by promoting transcription of the HBV core protein and by interacting with other AUBPs, i.e., HNRNPC and PTPB1, thus protecting viral RNA from degradation/splicing [94]. 

## 4. Modulation of Immune Responses by AUBPs in Inflammation and Cancer

More than 70% of HCC develops in an inflammatory environment [95], which fosters the recruitment of immune cells, uncontrolled hepatocyte proliferation, the accumulation of mutations, and malignant transformation that characterizes hepatocarcinogenesis [96]. Immune cell infiltration and deregulated cytokine production importantly contribute to the malignant transformation of hepatic cells [97]. The balance between pro- and anti-inflammatory responses is tightly controlled by different subsets of inflammatory mediators produced by different immune cells residing within distinct and specific liver compartments [98]. The constant release of proinflammatory cytokines favors the development of immunopathological niches (composed of neutrophils, macrophages, distinct subpopulations of T and B cells, etc.) that trigger chronic inflammation, hepatocyte damage, and fibrosis. These immunopathological niches are also involved in cancer development and chemoresistance, which are tightly associated with immune dysfunction and the gain of immune tolerance by malignant cells [98,99,100,101]. The sustained expression and release of these proinflammatory cytokines, such as TNFα, IL-1β, and IL-6, by immune cells in hepatic immunological niches, are tightly regulated by several AUBPs, including HuR, TTP, YB-1, TIA1, and KSRP, as previously mentioned [74]. 

In HepG2 hepatoma cells, APOBEC3B-induced HuR translocation into the cytoplasm has been reported to stabilize the mRNA of IL-6, a cytokine that regulates the balance between Th17 and Tregs cells [102,103]. Several studies have further highlighted the HuR-dependent modulation of T cell activation and polarization in physiological conditions and inflammatory diseases [75,104,105,106,107,108]. In line with such HuR-dependent regulatory mechanisms of T cell activation, destabilization of the transcription factor *Gata3* and Th2-specific cytokines (*Il2* and *Il13*) has been observed in CD4+ T cells isolated from HuR knockout mice, suggesting a decreased ability of T cell for Th2 polarization after activation [109]. 

The loss of TTP expression in cancer may also strongly influence antitumoral immune responses and evasion of the host immune system. Indeed, in colorectal cancer, TTP has been reported to destabilize the transcription of programmed death ligand 1 (PD-L1), a potent immunosuppressive protein upregulated in many cancers [110]. The restoration of TTP expression in mouse colon or gastric cancer cells further leads to decreased PD-L1 expression and increased antitumoral response [110,111]. In colon cancer cells, TTP also promotes the downregulation of *PTGS2*, a potent activator of prostaglandin E2 biosynthesis, which not only promotes cancer cell proliferation, but also impairs T cell activation, thus favoring immune escape [112,113,114]. However, in the liver, genetic ablation of TTP, specifically in the hepatocytes, decreases the inflammation and tumorigenesis induced by injection of a carcinogen (i.e., DEN) [86]. Whether this unexpected phenotype results from the inhibition of other TTP-dependent inflammatory processes (e.g., cytokine expression) altering the tumor microenvironment and fostering tumor progression remains currently unknown and needs further investigation [96,115]. 

YB-1 is another AUBP with a well-established oncogenic function in HCC [116]. YB-1 overexpression is typically linked to drug resistance and a worse prognosis, not only because it favors cancer cell proliferation, but also because it induces tumor immune evasion by impairing antitumoral responses through the modulation of cytokine expression. At the mechanistic level, YB-1, together with nucleolin, was shown to stabilize *Il2* expression in activated T cells by binding to the JNK-responsive regions in the 5’-UTR that share partial similarity with ARE sequences (i.e., CUUUA instead of AUUUA) [117]. Similarly, in LPS-stimulated dendritic cells, YB-1 has been reported to stabilize *Il6* transcripts [118]. 

TIA1 is also a crucial factor for immunity, since it balances the expression of inflammatory molecules (e.g., TNF-α and IL-1β) by modulating their translational initiation [17,119]. Supporting this function, macrophages from TIA1-depleted mice are more sensitive to the LPS-induced expression of TNF-α. Interestingly, the downregulation of the TIA1 cofactor TIAR in LPS-induced macrophages also leads to increased translation of *Il1b* transcripts [120]. Several studies have further highlighted TIA1 as a marker of the cytotoxic activities of immune cells, since it is a component of cytotoxic granules and is responsible for the apoptosis of targeted cells through mechanisms that are still unknown [121,122]. Consistent with this role of TIA1, increased infiltration of TIA1+ lymphocytes is associated with improved prognosis of patients diagnosed with colorectal cancer [122]. However, whether or not, in the liver, the TIA1/TIAR complex is essential for balancing immune responses, thus restraining chronic inflammation and hepatocarcinogenesis, remains to be investigated.

Finally, KSRP was recently reported to control T cell function. Indeed, KSRP genetic deletion in vivo leads to increased proliferation of CD4+ T cells and polarization toward Th2, thereby decreasing inflammation in mice following the induction of arthritis [123]. As for TIA1, whether KSRP regulates immune cell function during liver inflammation and/or carcinogenesis is still unknown.

Based on the studies described above, it is likely that deregulated AUBPs in liver inflammation and cancer deeply affect the efficiency of the immune responses to these disorders. However, studies focusing specifically on these aspects of liver diseases remain fragmentary, and in-depth analyses are required in the future to understand the pathophysiological role of specific AUBPs in hepatic immunity. 

## 5. AUBPs in the Hallmarks of Liver Cancer

Besides inflammation and evasion of the host’s immunity, HCC, as other cancers, arises from a set of events (i.e., mutations or nongenomic alterations) favoring the expression and/or activity of the genes involved in the promotion of cell survival migration and invasion, as well as the inhibition of genes triggering cell death. Many transcripts of these genes contain cis-regulatory sequences in their 3’-UTRs, which can be targeted by AUBPs for post-transcriptional regulation. As for genetic mutations, mRNA stabilization and promotion of the translation of oncogenes or, on the contrary, degradation of tumor-suppressive transcripts may steer cells into abnormal proliferation, inhibition of cell death, increased migration, and metabolic switch characterizing cancer cells [124]. It is therefore likely that alterations of the expression, activity, or intracellular localization of AUBPs represent key drivers of hepatic carcinogenesis. AUBPs such as TTP are constantly downregulated in HCC [86,125], while most of the other AUBPs are usually upregulated (i.e., HNRNPA1, PTBP1, RBM3, ILF3, or TIA1), a feature that often correlates with a worse survival prognosis [66,126,127,128,129]. In the following section, recent advances in our understanding of the role of AUBPs in key cancer hallmarks are discussed. 

### 5.1. Cell Cycle Regulation and Cell Proliferation 

ARE sequences are present in several transcripts of cell cycle regulators and other key drivers of cell proliferation, supporting the important role of AUBPs in carcinogenesis (Figure 5 and Table 1). HuR has been shown to control cell proliferation through various direct and indirect mechanisms in cancers, for example, by stabilizing the mRNAs of various cell cycle regulators such as those of *CCNA1*, *CCNB1,* and *CDK3* [130,131]. For instance, in a MAT1A knockout mouse model of HCC, translocation of HuR into the cytoplasm has been reported to stabilize *CCNA2* and *CCND1* transcripts [132]. Noncoding RNAs have also been shown to act as key cofactors of HuR to promote cell proliferation, as shown in the case of UFC1, a long-intergenic noncoding RNA overexpressed in human HCC, which enhances HuR stabilization of the *CTNNB1* mRNA, thereby stimulating the proliferation of HCC cells [133]. An acceleration of the G0/G1-to-S transition, leading to enhanced proliferation of hepatic cancer cells, is also triggered by HuR binding to the pleiotrophin downstream transcript (Ptn-dt), a lncRNA upregulated in mouse HCC [71]. Finally, HuR binding to the circRNA circBACH1 has been shown to favor HuR shuttling to the cytoplasm and the proliferation of Hep3B and HepG2 hepatic cancer cells [72]. 

Cell cycle progression and proliferation is also stimulated by other AUBPs through various mechanisms in hepatic cancer cells. PTBP1 has indeed been shown to increase the translation of *CCND3* in Hep3B cells [126], while ILF3 stabilizes *CCNE1* in Hep3B cells, thereby favoring G1/S transition [128]. YB-1 has also been reported to upregulate *CCNA2*, *CCNB1,* and *PCNA* and to downregulate *TP53* in HepG2 cells and in vivo in mouse livers [134]. On the other hand, destabilizing AUBPs have the ability to impair cell cycle progression. For instance, CUGBP1 silencing in HepG2 cells results in the downregulation of *CCNB1* and the upregulation of *CCND1,* leading to a G0/G1 blockade [135]. Similarly, the downregulation of *MDM2*, a P53 regulator destabilized by RBM38 binding, induces a G0/G1 blockade in HepG2 cells and suppresses tumor growth in mice xenographs [125].

Major signaling pathways leading to abnormal cell growth are also directly or indirectly modulated by specific AUBPs. This is the case, for example, for TTP, which inhibits the proliferation of HCC cell lines by reducing the half-life of the *MYC* mRNA and by blocking progression from the cell cycle S phase [86,136]. Additionally, by binding to the 3’-UTR of stearoyl-CoA desaturase (SCD), RBM3 induces the expression of a circRNA, SCD-circRNA 2, in HCC cells, which promotes proliferation, possibly through ERK phosphorylation [127]. Finally, based on gene set enrichment analysis, the high expression of HNRNPA1 in HBV-related HCC has been suggested to affect cell cycle progression and the WNT signaling pathway, pointing to HRNPA1 as a potent oncogene regulating the epidermal growth factor receptor (EGFR) signaling pathway [66].

### 5.2. Apoptosis

Cell death inhibition is an important feature of cancer cells and is a major cause of chemoresistance. In HCC, deregulated expression/activity of AUBPs severely affects the expression of the genes involved in apoptosis or the upstream signaling pathway regulating these processes (e.g., PI3K and MAPK; Figure 5 and Table 1). In particular, AUBPs exert tight control on (i) the intrinsic apoptotic pathway [137], which is triggered by DNA-damaging agents and involves mitochondrial events (e.g., loss of mitochondrial membrane potential and mitochondrial outer membrane permeabilization); (ii) extrinsic apoptosis, triggered by death receptors. However, AUBPs can also control other nonapoptotic cell death mechanisms, including ferroptosis, a specific cell death program triggered by iron overload and autophagy, or necroptosis, as it has been evidenced in nonliver tissues [138].

The deregulated expression and/or activity of AUBPs alters the intrinsic apoptosis pathway in cancers, mostly by impacting the expression of the Bcl-2 family members [124,139] or the upstream survival signaling pathways (e.g., PI3K/AKT and β-catenin signaling). Solid evidence supporting the role of AUBPs in the regulation of the intrinsic apoptosis pathway has been gathered in the case of TTP and HuR, but information about the other AUBPs still remains fragmentary. 

TTP has been demonstrated to directly bind to the 3’-UTR of the *HIF1A* mRNA and to promote its degradation in colon cancer, suggesting that TTP induction may represent a survival mechanism against prolonged hypoxia [140]. TTP silencing, likely by preventing the downregulation of specific TTP targets, including *MYC*, *IER3,* and *AKT1,* has been reported to equally restrain apoptosis induced by a MAPK inhibitor (PHA-781089) in hepatic cancer cell lines [141]. Consistent with these studies, TTP overexpression in HCC cells has been shown to be associated with the downregulation of *BCL2* in PLC/PRF/5 cells and *MYC* in Huh7 and HepG2 cells [86]. However, whether TTP directly controls the mRNA decay of these transcripts is still unclear and was not validated in vivo in mice with TTP specifically deleted in the liver, which unexpectedly presents a paradoxical phenotype with reduced tumor burden induced by the hepatic carcinogen DEN [86]. 

The oncogenic activity of HuR has been reported in various cancers [142], but the currently available literature indicates that in HCC, HuR can either promote or inhibit apoptosis depending on the cell model. Indeed, HuR has been shown to contribute to TIP30-induced apoptosis by directly stabilizing the *TP53* mRNA in HL7702 cells [143], and is also required for LKB1-dependent control of apoptosis in a MAT1A-deficient hepatic cell line [144]. LKB1 has been further shown to promote HuR cytoplasmic translocation, which, in turn, stabilizes the herpesvirus-associated ubiquitin-specific protease (HAUSP) mRNA, a nuclear ubiquitin-specific protease controlling the stability of P53 [144,145]. HuR has also been shown to be involved in Hedgehog-interacting protein antisense RNA-1 (H-AS1)-induced apoptosis by stabilizing the mRNA of *HHIP*, a negative regulator of the Hedgehog pathway, in Hep3B cells [146]. In contrast to these studies, HuR has been reported to inhibit arsenic trioxide (ATO)-induced apoptosis by preventing ATO-induced TGFβ/SMAD signaling. This effect appears to be mediated by an HuR-dependent increase in the stability of the TG-interacting factor (TGIF) mRNA [147]. In another study, β-catenin stabilization by HuR’s interaction with UFC1 has been further shown to inhibit apoptosis [133]. Finally, HuR is also able to activate other prosurvival pathways in HCC cells, including HER2 signaling, upon HBV infection [148]. Together, these findings indicate that, depending on the cell model or the proapoptotic stimuli, HuR can exert both a pro- and apoptotic function.

As mentioned, the role of other AUBPs on intrinsic apoptosis pathways is currently poorly understood. Nevertheless, one study demonstrated that TIA1 directly inhibits the expression of the tumor suppressor *IGFBP3* in hepatic cancer cells [149]. Although the mechanistic link with apoptosis was not clearly established in this study, IGFBP3 is a potent inhibitor of the prosurvival insulin-like growth factor (IGF) signaling, and its overexpression induces the intrinsic apoptotic pathway through various mechanisms (e.g., through P53 signaling) [150,151]. CUGBP1 and CUGBP2 are also potentially important for the regulation of apoptotic processes and, despite some similarities, these two proteins seem to exert opposite functions in HCC, although their mechanisms of action remain to be defined. Indeed, CUGBP1 overexpression can prevent caspase-dependent apoptosis induced by a piperazine derivative, BK10007S, in HepG2 cells [152], while CUGBP2 may contribute to brucein D-induced apoptosis in hepatic cancer cell lines [67]. ILF3, another AUBP that is strongly upregulated in HCC, and directly binds and stabilizes the *PARP1* mRNA in HCC cells [153]. The depletion of ILF3 in HCC cells sensitizes cancer cells to a PARP inhibitor (i.e., Olaparib) and to a DNA-damaging agent (i.e., 10-hydroxycamptotecin). Some AUBPs may also exert an indirect regulatory function in apoptosis. For instance, AUF1 plays an apoptotic function in HCC by impairing the maturation of the proapoptotic miRNA, miR-122, through the direct regulation of Dicer [154]. Although the precise mechanisms need further clarification, miR-122 inhibits TLR4 [155], β-catenin [156], and IGF1R signaling [157], thereby supporting the indirect proapoptotic function of AUF1. Of note, AUF1 has also been reported to directly bind to the mRNA of MAT1A and to stimulate its decay [158]. MAT1A is a potent apoptosis inducer, as its forced expression in human HCC cells increases the expression of various proapoptotic genes (e.g., *ARH1*, *FRZB*, and *PP2A*), while reducing the activity of prosurvival pathways (i.e., ERK and PI3K/AKT signaling) and hepatic tumorigenesis in mice [159]. Finally, the P53 pathway is also regulated by AUBPs, such as RBM38, which is downregulated in HCC. Interestingly, RBM38 overexpression in HCC cells promotes apoptosis in a p53-dependent manner by destabilizing the *MDM2* transcript through direct interaction with an AU-rich sequence within its 3’-UTR [125]. 

AUBPs also play an important regulatory role in extrinsic apoptosis by directly controlling the expression of death receptors, their ligands, and downstream signaling proteins (e.g., caspase-8) [124]. Little information is available regarding the control of extrinsic apoptosis by AUBPs in HCC, but HuR has been shown to downregulate the surface expression of Fas in HCC cells, and its silencing re-sensitizes HCC cells to FasL-induced apoptosis [160]. Although preliminary, these findings suggest that targeting the expression/activity of specific AUBPs may restore the response of cancer cells to physiological death stimuli and may potentially counteract the immune escape of cancer cells. Ferroptosis, an iron-dependent nonapoptotic cell death process impaired in HCC [161], is also sensitive to alterations in the expression/activity of AUBPs in the liver. Indeed, TTP has been shown to exhibit protective activity against ferroptosis in hepatic stellate cells, through an inhibition of autophagy mediated by TTP binding to the 3’-UTR of the autophagy-related 16-like 1 (*ATG16L1*) mRNA, which promotes its degradation [162]. Interestingly, the overexpression of TTP in HSCs prevents sorafenib- and errastin-induced ferroptosis and promotes liver fibrosis induced by a bile duct ligation. As mentioned earlier, HuR, in contrast, promotes the ferroptosis of HSCs and thus prevents liver fibrosis by increasing autophagy through the stabilization of the mRNA of *BECN1* [79]. A link between HuR and autophagy has been further observed in another study, where a HuR-dependent stabilization of *ATG5*, *ATG12,* and *ATG16* mRNAs was observed in LX-02 and Hep3B cells [163]. 

### 5.3. Migration/Invasion/Metastasis

The migration/invasion capacity and the ability of HCC cells to form intra- and extrahepatic metastases are major determinants of patient survival. AUBPs, either alone or in combination with the action of ncRNAs, significantly impact this cancer hallmark (Figure 5 and Table 1). For instance, interaction of the HuR mRNA with the lncRNA AK058003 has been shown to destabilize γ-Synuclein transcripts and to inhibit their expression, thereby decreasing the migration/invasion properties of HCC cancer cells both in vitro and in vivo [164]. Furthermore, the upregulation of HuR by the HBx viral protein has been reported to stabilize HER2 expression and to confer a higher migratory capacity in Hep3Bx hepatic cancer cells [148]. The overexpression of TTP in hepatoma cells has been shown to decrease cell migration, thus supporting the potential tumor-suppressive function for this AUBP [86], whereas RBM38 has been reported to downregulate the migration/invasion of HepG2 cells by affecting the stability of the *MDM2* mRNA and p53 signaling [125]. YB-1 has been demonstrated not only to prime HCC initiation via the Wnt/β-catenin pathway, but also to promote Huh7 HCC cell migration by triggering epithelial-to-mesenchymal transition [134]. In this regard, prostaglandin E2 has been shown to promote HCC invasion by increasing YB-1 expression in HCC through Src-, EGFR-, and p44/42 MAPK-dependent signaling [165]. Furthermore, the lncRNA AWPPH has been suggested to increase, in vitro, the migration capacities of SMMC-772 and HCCLM3 cells and to favor in vivo metastasis formation in mouse xenografts by stimulating Snail1 expression and translation via YB-1 activation [63]. Other lncRNAs have also been suggested to affect the migratory and invasion properties of HCC cells through the action of AUBPs. For example, the lncRNA ANCR upregulates HNRNPA1, which, in turn, promotes the in vitro migration/invasion of Hep3B, Huh7, and HepG2 cells, as well as the metastasis of HCC in a xenograft model [166]. On the contrary, the lncRNA MIR22HG inhibits HCC cell migration/invasion by directly interacting with the HuR protein and promoting its translocation from the cytosol to the nucleus, where stabilization of the oncogenic transcript by HUR (i.e, β-catenin) is less effective [70].

### 5.4. Angiogenesis

Angiogenesis is an important cancer hallmark required by solid tumors to progress to later stages and aggressiveness by allowing an appropriate blood supply of nutrients, oxygen, and building blocks during the anarchic proliferation of cancerous cells [167]. The overexpression of TTP in PLC/PRF/5 and HepG2 cells was shown to downregulate the expression of *VEGFA*, a key proangiogenic factor, and low TTP expression is correlated with higher vascular invasion in HCC patients [86]. The hypoxia-induced factor 1α (HIF1α), another major proangiogenic factor regulating the expression of *VEGFA* and correlating with poor outcome in HCC, also contains ARE sequences [168]. When hypoxia is induced by CoCl2 in HeLa cells, both HuR and PTBP1 trigger the translation of the *HIF1A* mRNA, without affecting its abundance, by binding to its 5’-UTR and 3’-UTR, respectively [169]. Intriguingly, while this direct targeting of *HIF1A* translation has not been confirmed in hepatic cells, an indirect relationship between HuR and HIF1α has been established in HCC. Indeed, HuR has been shown to suppress the maturation of miR-199a, a negative regulator of HIF1α, in Hep3B cells, resulting in the upregulation of the targets of HIF1α [170,171]. 

### 5.5. Metabolic Reprogramming

The development of liver tumors is highly dependent on metabolic reprogramming occurring already in the precancerous stages and during cancer progression. This reprogramming includes, in particular, increased aerobic glycolysis, induction of the pentose phosphate pathway, increased glutamine catabolism, enhanced amino acid and lipid metabolism, mitochondrial biogenesis, macromolecular synthesis, and redox homeostasis [172]. Alterations of these metabolic processes represent key drivers of tumorigenesis and malignancy in the liver [173], and several AUBPs have been linked to these metabolic changes in HCC and other cancers. 

Alterations of the lipid metabolism in hepatic cancers have been shown, for example, to be dependent on the activity of TTP. Indeed, hepatic lipid accumulation induced by the carcinogen DEN is abrogated in mice bearing a hepatocyte-specific deletion of TTP, and the ratio of hepatic saturated and mono-unsaturated lipids in these mice is different compared to wild-type mice [86]. TTP has also been reported to indirectly affect the hepatic lipid and glucose metabolism by promoting insulin resistance in mice fed an obesogenic diet through targeting of *Fgf21* [12]. Finally, TTP has also been shown to regulate glycolytic levels in hepatic cancer cells by lowering the expression of *MYC*, an important regulator of glycolysis [86].

As mentioned in the previous section, HuR regulates the expression of HIF1α in HCC [169]. Besides its role in hypoxia and angiogenesis, HIF1α is also a potent activator of key glycolytic genes, among which are hexokinase 2 (HK2) and pyruvate kinase M2 (PK-M2; a tumor-specific isoform of PK), as demonstrated in Hep3B cells [170]. Although direct evidence of glycolysis regulation by HuR is not available, it is highly probable that the HuR–HIF1α axis is relevant in glycolytic switches occurring in HCC, but further investigation is required to test this hypothesis. Glycolysis has also been shown to be significantly affected in HCCLM3 and Hep3B cells following miR-374b-mediated downregulation of the AUBP hnRNPA1, which induces the expression of the rate-limiting enzyme PKM2 [174]. Of note, hnRNPA1 has also been reported to directly regulate the levels of cytochrome P450 2A6 (*CYP2A6*) in HepG2 cells, thus highlighting its broad effect on xenobiotics metabolism and cellular stress as well [175]. 

Finally, the two AUBPs AUF1 and HuR have been found to deeply impact the amino acid metabolism in the liver of rats treated with the carcinogen DEN. The levels of S-adenosylmethionine (SAM) were indeed decreased by the activity of these two AUBPs, which regulate the gene expression of methionine adenosyltransferase 1a and 2a (*Mat1a* and *Mat2a*) [176]. Of importance, a metabolic switch from *Mat1a* to *Mat2a* expression, which translates into the downregulation of SAM levels, is a well-known metabolic reprogramming event favoring HCC progression. 

## 6. AUBPs as Biomarkers and Potential Therapeutic Targets

### 6.1. AUBPs as Biomarkers for Chronic Liver Diseases and HCC 

The deregulated expression/activity of AUBPs may represent an important biomarker for chronic liver diseases and HCC, and thus may be a potential indicator of a patient’s prognosis and sensitivity to the currently available therapeutic strategies (e.g., sorafenib and doxorubicin). However, very few studies have been conducted to evaluate the potential of AUBPs as biomarkers of liver diseases and cancers. Indeed, although the deregulated expression patterns of AUBPs are observed in patient cohorts with liver cancers, there is little information on whether these alterations are associated with particular etiological factors or specific HCC subtypes. For instance, HuR and TTP are typically up- and downregulated, respectively, in HCC, but also in the early stages of liver disorders caused by HBV infection [148,177]. It is thus unclear whether HuR/TTP up-/downregulation is a specific and independent feature of HCC, or of HBV infection. Future in-depth analyses should ideally combine measures of AUBP expression, other more conventional HCC markers (e.g., GPC3, HSP70, and GS) [178], and etiological factors to refine the relevance of AUBPs as biomarkers and their potential use for patient diagnosis. Moreover, the activity of AUBPs, and not solely their expression, need to be considered. Since the localization of AUBPs in the cytoplasm versus the nucleus appears to be a direct read-out of their activity, immunohistological analyses of the localization of specific AUBPs in the cytoplasm/nucleus should also be considered as a prognostic tool. In support of this concept, the cytoplasmic localization of HuR is associated with poor survival of ICC patients receiving adjuvant gemcitabine-based chemotherapy [179].

Finally, it will be important to determine whether abnormal AUBP levels can be detected in liquid biopsies from patients with chronic liver diseases or HCC, since pilot studies have suggested the presence of AUBPs in extracellular vesicles (e.g., exosomes) [180,181]. Very few circulating markers are currently used for the diagnosis of chronic liver diseases and HCC, among which, alpha-fetoprotein (AFP) is frequently used in clinics for HCC diagnosis and for evaluating tumor recurrence [182]. However, AFP is not always reliable for predicting patient outcomes, tumor differentiation, or malignancy [182]. Moreover, it is unclear whether AFP is a reliable serum marker for HCC development in noncirrhotic patients. Assessing the signature and/or localization of AUBPs in the solid and/or liquid biopsies of patients may provide additional valuable tools to detect the severe stages of liver inflammation and fibrosis, as well as to grade HCC and to evaluate patients’ prognosis. 

### 6.2. Therapeutic Targeting of AUBPs and SG Formation in Liver Diseases and HCC

The involvement of AUBPs in typical cancer-related cellular processes and pathways associated with HCC development suggests the potential for novel therapeutic approaches aimed at targeting the expression and/or activity of these proteins. Moreover, targeting these proteins in chronic liver diseases such as NAFLD/NASH could be used as a preventive approach to limit the severity of these diseases and their progression toward cancer. RNA-binding proteins have been previously qualified as “undruggable” due to the lack of a binding pocket on active sites that can be targeted by small molecules. However, high-throughput methods have allowed the identification of potential molecules affecting their activity. Moreover, strategies aimed at reducing/increasing their expressions/activities have been identified and display potent antitumoral properties. Currently, most therapeutic strategies aim to inhibit HuR expression/activity or restore TTP expression, while very few approaches have been developed for the other AUBPs.

HuR represents an appealing therapeutic target due to its increased expression/activity in cancer cells and its prosurvival functions. Targeting HuR may also re-sensitize hepatic cancer cells to other existing therapeutic approaches (i.e., chemotherapy and sorafenib). Therefore, several strategies aimed at inhibiting HuR function have been developed, including the inhibition of (i) its translocation to the cytosol, (ii) its interaction with ARE sequences, and (iii) its expression [183]. High-throughput screening approaches have identified several molecules with the capacity to inhibit HuR functions [184]. Polyketides purified from microbial extracts or plants (i.e., okicenone, dehydromutacin, and MS-444) display specific HuR inhibitory properties. Among them, MS-444 represents an interesting candidate due to its antitumor properties in various cancers (e.g., colorectal cancer, pancreatic cancer, and malignant gliomas) [185,186]. MS-444 inhibits HuR homodimerization and prevents its cytoplasmic export, thereby reducing the stability of its mRNA targets. Moreover, the antitumor effect of MS-444 has also been observed in vivo in a mouse model of inflammatory bowel disease, as well as in a genetic mouse model of familial adenomatous polyposis (i.e., APC^Min^ mice). However, the therapeutic potential of this molecule in chronic liver diseases and HCC has not been evaluated. Targeting the AMPK signaling pathway may also affect HuR translocation. 5-Aminoimidazole-4-carboxamide ribonucleotide (AICAR), a well-known activator of AMPK, induces the nuclear retention of HuR in muscle C2C12 cells [187]. However, this effect is cell type-dependent and recent findings in HepG2 cells indicate that AICAR promotes, on the contrary, HuR cytoplasmic export [188]. The cytoskeletal inhibitors latrunculin A and blebbistatin exert antitumoral properties in HCC cells by interfering with HuR cytoplasmic export [189]. Similarly, the cantharidin analog N-benzylcantharidinamide reduces Hep3B invasion by blocking the translocation of HuR to the cytosol [190]. Several molecules that impair the HuR/ARE interaction have also been identified, including dihydrotanshinone (DHTS) [191] or CMLD-2, a coumarin-derived molecule, which impairs the HuR/ARE interaction in pancreatic, colon, and lung cancers [191]. Interestingly, DHTS has antitumoral properties in HCC cell lines [192], but it is unclear whether this effect is strictly associated with an impairment in HuR function. Finally, different approaches aimed at reducing HuR expression in cancer cells have been proposed and include the delivery of specific siRNAs through various carriers, including nanoparticle-based delivery, liposome-based nanoparticles, poly amidoamine nanoparticles, or DNA dendrimer nanocarriers [193]. Interestingly, these carriers can be modified to specifically target cancer cells as demonstrated in lung cancer [194]. However, the efficiency of such approaches remains to be established in chronic liver diseases and HCC. 

TTP loss in human HCC occurs mostly through the DNA methylation of its promoter [136], and its expression can be restored by DNA-demethylating agents, such as decitabine [141]. This approach sensitizes HCC cells to MK2 inhibitor-induced apoptosis and thus may also re-sensitize cancer cells to chemotherapy, as suggested in other cancers [195]. In addition, other substances of natural origin, such as resveratrol or green tea extracts, also display an ability to increase TTP expression in colon cancer cells and in the liver of rats fed a high-fructose diet, respectively [196,197]. These studies suggest that polyphenols may represent appealing therapeutic molecules for the treatment of chronic liver diseases and HCC. However, as mentioned earlier, TTP loss in the mouse liver reduces tumor burden in vivo, despite obvious tumor-suppressive properties in vitro [86]. Moreover, in HSCs, TTP inhibits ferroptosis in HSCs and thus promotes liver fibrosis [162]. These paradoxical results suggest a double function of TTP during liver carcinogenesis and thus highlights the need for caution regarding the use of molecules for restoring TTP expression in the whole liver.

Very few studies have attempted to evaluate the potential of targeting other AUBPs for therapeutic purposes. Silencing of HNRNPA1 or NCL in HCC cells using small RNA fragments (i.e., Aptamer BC15 for HNRNPA1 or AS1411 and AGRO100 for NCL) results in a potent reduction of cancer cell growth [198,199]. Interestingly, the conjugation of AS1411 with doxorubicin (AS1411–DOX adduct) improves the delivery of DOX to hepatic cancer cells and thus may reduce the potential side effects of DOX [200]. ILF3 may also represent an appealing target, as its depletion in HCC cells improves the antitumoral effect of the PARP inhibitor olaparib and of the DNA-damaging agent 10-hydroxycamptotecin [153]. However, despite these promising in vitro findings, the relevance of such strategies remains to be demonstrated in vivo.

## 7. Conclusions

AUBPs are an important family of proteins controlling the cellular transcriptome. By targeting the transcripts involved in the control of inflammation, immune cell functions, cell growth, cell death, and metabolism, AUBPs are key players modulating the development of numerous pathologies, including chronic liver inflammation/fibrosis and cancer. These RNA-binding proteins are subject to self-regulation, as well as competition and synergy with each other. In addition, many other cellular factors such as miRNAs and lncRNAs compete or synergize with AUBPs in different processes, thereby drastically increasing the complexity of these regulatory networks. Currently, most studies have focused on HuR and TTP, highlighting the pathophysiological importance of these post-transcriptional regulators. Further studies are now required to investigate the specific roles and functions of other members of this family in cell physiology and in diseases. By targeting multiple transcripts and processes in cells, AUBPs represent important signaling nodes relevant for therapeutic targeting, since they should be able to kill many birds with one stone. This is particularly important in multifactorial and heterogenous diseases such as NASH, fibrosis, and liver cancers. However, despite encouraging in vitro and in vivo studies, our understanding of the roles and functions of AUBPs and their cofactors in the liver remains fragmentary, especially in the case of less common hepatic cancers, such as ICC. Further extensive studies are now required to deepen our knowledge about AUBPs and to confirm the great potential that targeting of these proteins holds for future therapies of liver diseases and cancers. 

## Figures and Tables

**Figure 1 ijms-21-06648-f001:**
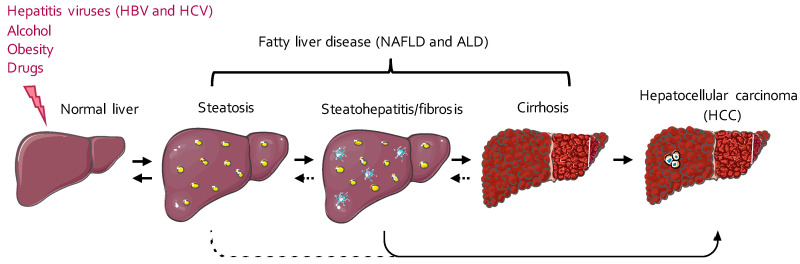
The spectrum of the different stages of fatty liver disease (FLD) and its progression toward hepatocellular carcinoma (HCC). The first stage of FLD usually consists of an aberrant accumulation of fat in the hepatocytes, under the form of cytoplasmic lipid droplets, a pathological situation called steatosis. Hepatic steatosis can then progress to steatohepatitis, a chronic inflammatory state often accompanied by the development of fibrotic scars replacing the parenchymal hepatic tissue. Extended fibrosis and the development of undifferentiated regenerating hepatocyte nodules further characterize the progression of steatohepatitis and fibrosis to cirrhosis. Cirrhosis represents an important risk factor for the progression of FLD toward the development of hepatocellular carcinoma (HCC), which may also occur in noncirrhotic livers displaying only inflammation/fibrosis. ALD, alcoholic liver disease; HBV, hepatitis B virus; HCV, hepatitis C virus; NAFLD, nonalcoholic fatty liver disease.

**Figure 2 ijms-21-06648-f002:**
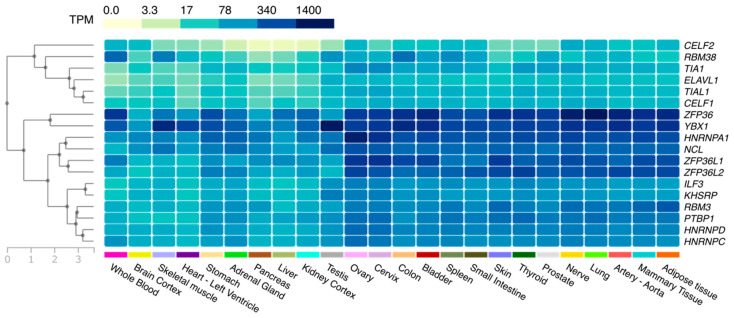
Tissue distribution of AU-rich element-binding proteins (AUBPs). A heat map representing the hierarchical clustering (left) of the mean relative expression (transcripts per million (TPM)) of AUBPs in different tissues and organs. The data used for the analyses were obtained from the Genotype-Tissue Expression (GTEx) project’s portal (https://www.gtexportal.org/home/) on 24 June 2020.

**Figure 3 ijms-21-06648-f003:**
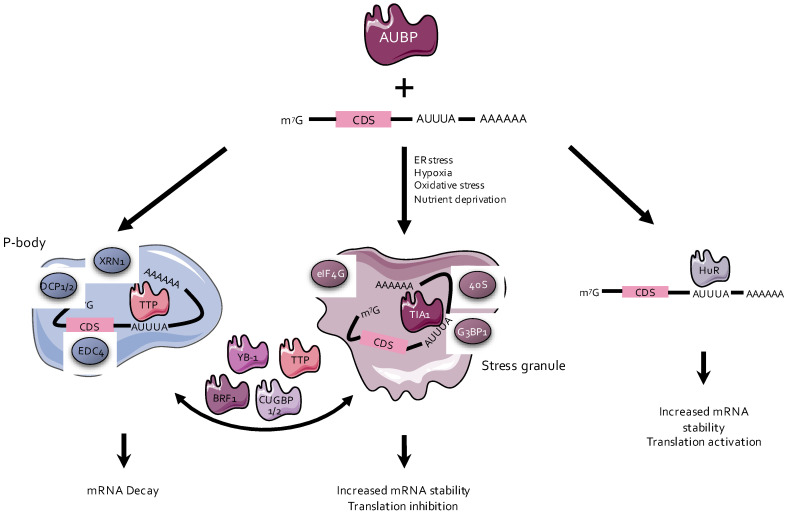
Functions of AU-rich elements-binding proteins (AUBPs). AUBPs associated with processing bodies (P-bodies), such as tristetraprolin (TTP), bind to RNAs and promote their degradation, while AUBPs associated with stress granules (SGs), e.g., T-cell-restricted intracellular antigen-1 (TIA1), inhibit translation and preserve their bound mRNA from degradation. AUBPs can also transfer target mRNA from P-bodies or SGs, or vice versa, to change the fate of targeted transcripts. Other AUBPs, exemplified by Hu-antigen R (HuR), may also bind transcripts, increase their stability, and/or activate their translation. BRF1, butyrate response factor 1; CDS, coding sequence; CUGBP1/2, CUG triplet repeat RNA-binding protein ½; ER, endoplasmic reticulum; YB-1, Y-box-binding protein 1.

**Figure 4 ijms-21-06648-f004:**
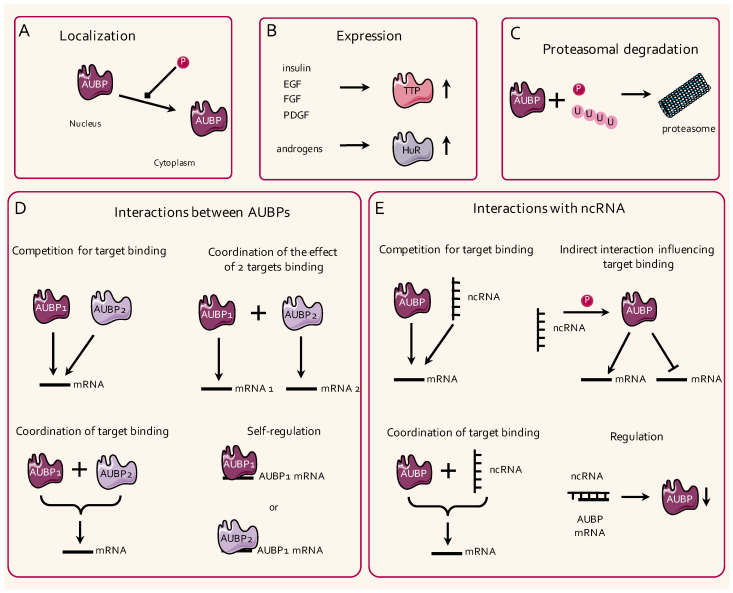
Regulatory mechanisms of the activity of AUBPs. (**A**) The localization and shuttling of AUBPs between the nucleus and the cytoplasm can be regulated by phosphorylation events. (**B**) The expression of AUBPs, such as TTP or HuR, is regulated by growth factors (e.g., insulin, epidermal growth factor (EGF), fibroblast growth factor (FGF), and platelet-derived growth factor (PDGF)) and androgens, respectively. (**C**) AUBPs can be targeted for proteasomal degradation by phosphorylation and/or ubiquitination. (**D**) AUBPs can compete for binding to the same mRNA (upper left panel) or can influence their respective binding to different or to the same target mRNAs (upper right and lower left panels). AUBPs can finally regulate the lifetime of their own mRNAs and of the RNAs of other AUBPs (lower right panel). (**E**) AUBPs and non-coding RNAs (ncRNAs) can compete or coordinate each other’s binding to a target mRNA (upper and lower left panels). ncRNAs can also indirectly influence the binding of AUBPs to their respective targets through intermediate mechanisms (e.g., phosphorylation events; upper right panel). Finally, ncRNAs can bind to the mRNAs of AUBPs and can regulate their protein expression levels (lower right panel).

**Figure 5 ijms-21-06648-f005:**
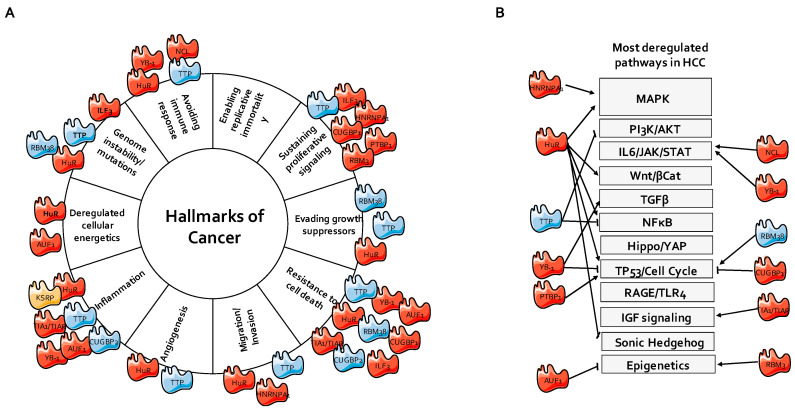
Network of AUBPs involved in liver cancer. (**A**) AUBPs regulating different hallmarks of cancer. (**B**) AUBPs regulating classical signaling pathways deregulated in HCC. The pointed arrows demonstrate positive regulation, while the blunt arrows demonstrate an inhibitory interaction. Red color shows AUBPs upregulated in HCC, blue color shows AUBPs downregulated in HCC patients compared to healthy patients, while orange color shows AUBPs, whose expression is unchanged, based on the Gene Expression Omnibus (GEO) dataset GSE89377. HCC, hepatocellular carcinoma; IGF, insulin-like growth factor; JAK, Janus kinase; MAPK, mitogen-activated protein kinase; NFκB, nuclear factor kappa-light-chain-enhancer of activated B cells; PI3K, phosphoinositide 3-kinase; STAT, signal transducer and activator of transcription; TGFβ, tumor growth factor β; TLR, Toll-like receptor; YAP, yes-associated protein, RAGE, Receptor for Advanced Glycation Endproducts.

**Table 1 ijms-21-06648-t001:** The validated targets of AUBPs controlling various processes in hepatic cells. BMDMs, bone marrow-derived macrophages; BMSCs, bone marrow-derived stem cells; HCC, hepatocellular carcinoma; HSCs, hepatic stellate cells; PMID, PubMed identifier.

AUBP	HCC	Target	Interaction	Model/Cell Type	Process	PMID
HuR	Up	*ACTA1*	Direct	HSCs	Fibrosis, migration	22576182
		*Col1a1*	Unclear	HSCs	Fibrosis	22576182
		*MMP9*	Direct	HSCs	Fibrosis, migration	22576182
		*MCP1*	Direct	HSCs	Liver Fibrosis	22576182
		*CCND1*	Direct	HSCs	Fibrosis, proliferation	22576182
		*Tgfb*	Direct	HSCs	Fibrosis	22576182
		*CCNB1*	Direct	HSCs	Fibrosis, proliferation	22576182
		*S1PR3*	Direct	BMSCs	Fibrosis, migration	27543493
		*CNR1*	Direct	BMDMs	BMDM recruitment	31495934
		*HMOX1*	Unclear	Hepatocytes	Oxidative stress	31879990
		*TP53*	Direct	HCC cells	Proliferation, apoptosis	18519672
		*FAS*	Direct	HCC cells	Apoptosis	25678597
		*MAT2A*	Direct	HCC cells	Proliferation, differentiation	20102719
		*Ptn-dt*	Direct	HCC cells	Cell proliferation	30643194
		*BECN1*	Direct	HSCs	Autophagy/Ferroptosis	30081711
		*ATG5*	Direct	HCC cells	Autophagy	30602494
		*ATG12*	Direct	HCC cells	Autophagy	30602494
		*ATG16*	Direct	HCC cells	Autophagy	30602494
		*HIF1A*	Direct	HCC cells	Hypoxia/Angiogenesis	30083257
		*IL6*	Direct	HCC cells	Inflammation	28646470
		*CCNA2*	Direct	Hepatocytes	Proliferation	16831604
		*CCND1*	Direct	Hepatocytes	Proliferation	16831604
		*ETS1*	Direct	HCC cells	Proliferation	31438961
		*UFC1*	Direct	HCC cells	Proliferation	25449213
		*HAUSP*	Direct	HCC cells	P53 signaling	20815019
		*HHIP*	Direct	HCC cells	Proliferation, apoptosis, migration	31604528
		*TGIF*	Direct	HCC cells	Apoptosis	21649584
		*CTNNB1*	Direct	HCC cells	Apoptosis	25449213
TTP	Down	*PTGS2*	Unclear	Hepatocytes	Inflammation	26876787
		*DUSP1*	Unclear	Hepatocytes	Proliferation	26876787
		*FOS*	Unclear	Hepatocytes	Proliferation	26876787
		*MYC*	Direct	Hepatocytes	Proliferation	20038433
		*MMP2*	Direct	HSCs	Migration	30226813
		*TNFa*	Direct	HSCs	Inflammation	30226813
		*FGF21*	Direct	Hepatocytes	Insulin sensitivity	29997282
		*VEGFA*	Unclear	HCC cells	Angiogenesis	31717307
		*ATG16L1*	Direct	HSCs	Autophagy, ferroptosis	31679460
		*IER3*	Unclear	HCC cells	Apoptosis	27619201
		*AKT1*	Unclear	HCC cells	Apoptosis	27619201
AUF1	Up	*MAT1A*	Direct	HCC cells	Proliferation, differentiation	20102719
		*Dicer*	Unclear	HCC cells	MicroRNA maturation	29599909
TIA1	Up	*IGFBP3*	Direct	HCC cells	Apoptosis	20599318
		*MFF*	Direct	HCC cells	Mitochondrial metabolism	27612012
YB-1	Up	*SMAD2*	Direct	HSCs	TGFβ signaling	28153731
		*EGFR*	Unclear	HCC cells	Proliferation	24378923
		*CCNA*	Unclear	HCC cells	Proliferation	27911878
		*CCNB1*	Unclear	HCC cells	Proliferation	27911878
		*PCNA*	Unclear	HCC cells	Proliferation	27911878
		*TP53*	Unclear	HCC cells	Proliferation, apoptosis	27911878
		*SNAI1*	Direct	HCC cells	Migration	28428004
KSRP	Unclear	*Per2*	Unclear	Hepatocytes	Metabolism	25514904
PTBP1	Unclear	*CCND3*	Direct	HCC cells	Proliferation	31301177
RBM38	Down	*MDM2*	Direct	HCC cells	Proliferation, apoptosis	30176896
RBM3	Down	*SCD-circRNA 2*	Direct	HCC cells	Proliferation	31235426
CUGBP1	Up	*CCNB1*	Unclear	HCC cells	Proliferation	24502807
		*CCND1*	Unclear	HCC cells	Proliferation	24502807
HNRNPA1	Up	*PKM2*	Unclear	HCC cells	Metabolism	31106002
		*CYP2A6*	Direct	HCC cells	Metabolism	15155834
ILF3	Up	*PARP1*	Direct	HCC cells	Apoptosis	28487110
		*CCNE1*	Direct	HCC cells	Proliferation	25399696

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
