# Peer review of "mRNA Post-Transcriptional Regulation by AU-Rich Element-Binding Proteins in Liver Inflammation and Cancer"

_ijms, 2020, doi:10.3390/ijms21186648_

Round 1
Reviewer 1 Report
AU-rich elements-binding proteins (AUBPs) are an important family of proteins controlling the cellular transcriptome. By targeting transcripts involved in the control of inflammation, immune cell functions, cell growth, cell death and metabolism, AUBPs are key players modulating the development of numerous pathologies including chronic liver inflammation/fibrosis and cancer. In this manuscript, the authors provide a detailed discussion on the roles and functions of AUBPs in liver diseases and cancer. The manuscript is well written and the illustration is presented in a good quality. This will provide interesting information for the reader of the journal. However, there are still many grammatical and syntax errors in the article. So I think the manuscript can be published after grammar and language check.Author Response
We thank the reviewer for her/his positive appreciation of our manuscript. As suggested the manuscript has now been edited to correct English grammar/syntax errors by the English language proof-reading service provided by IJMS.
Reviewer 2 Report
In the present review the authors summarize available literature on the role of mRNA post-transcriptional regulation by AU-rich elements-binding proteins in liver inflammation and liver cancer. By including in total 200 references the authors provide a very complete overview on available information and provide a well balanced discussion on the topic. The present review will be of high value for many readers since the important role of these proteins in the development of chronic metabolic and inflammatory fatty diseases as well as in HCC has been recognized only recently and similar reviews are lacking. To my best opinion the readers will benefit from different nicely layouted figures further summarizing the (complex) topic. I really have no further suggestion, it is a well done manuscript written by obvious experts in the field. If you really need a suggestion: “The authors might consider to included information on cholangiocarcinoma as well as on benign liver lesions as non-HCC liver tumors.”
The authors are congratulated to a very nice review including 200 references and a nice panel of figures.
Author Response
We thank the reviewer for her/his positive appreciation of our manuscript. The manuscript has now been edited to correct English grammar/syntax errors by the English language proof-reading service provided by IJMS.
As suggested by the reviewer, we also performed an extensive literature search about the role of AUBPs in non-HCC liver cancers i.e., intrahepatic cholangiocarcinoma (ICC) or benign liver tumors (focal nodular hyperplasia or hepatocellular adenoma). Unfortunately, information on this topic is very scarce and we found only one study assessing the role of the AUBP HuR in cholangiocarcinoma [1]. This study was already described in the subsection 6.1: “AUBPs as biomarkers for chronic liver diseases and HCC” (l. 1957) but we erroneously mentioned HCC instead of ICC when discussing this study. We apologize for this typo mistake, which has been corrected now. We also indicated now in our introduction (l. 89-91) and conclusions (l. 2285-2286), that very little very is known about the role of AUBPs in non-HCC liver cancer and that further studies are required to gain insights into AUBPs functions in these different types of liver cancers.
[1] Toyota K, Murakami Y, Kondo N, Uemura K, Nakagawa N, Takahashi S, Sueda T. Cytoplasmic Hu-Antigen R (HuR) Expression is Associated with Poor Survival in Patients with Surgically Resected Cholangiocarcinoma Treated with Adjuvant Gemcitabine-Based Chemotherapy. Ann Surg Oncol 2018;25(5):1202-10.